# Momentum via Primal Averaging: Theoretical Insights and Learning Rate Schedules for Non-Convex Optimization

## Abstract

Momentum methods are now used pervasively within the machine learning community for training non-convex models such as deep neural networks. Empirically, they outperform traditional stochastic gradient descent (SGD) approaches. In this work we develop an Lyapunov analysis of SGD with momentum (SGD+M), by utilizing an equivalent rewriting of the method known as the stochastic primal averaging (SPA) form. This analysis is tight enough to give precise insights into when SGD+M may outperform SGD, and what hyperparameter schedules will work and why. Surprisingly, we show that the commonly used stage-wise schedule doesn't make sense in SPA form, and discuss how to fix it. Our theory suggests that momentum is only useful at the early stages of training, and we verify this empirically by showing that dropping momentum after one epoch results in no loss of final test accuracy on CIFAR-10 and ImageNet training.

## 1 Introduction

Heavy ball methods have a long history dating back to the work of Polyak (1964). More recently, the stochastic heavy ball method, also known as stochastic gradient descent with momentum (SGD+M), has become a standard for deep learning practitioners since it was observed that momentum significantly helps on common computer vision problems (Sutskever et al., 2013).

In this work we provide an analysis of SGD+M for non-convex problems that is much tighter than past approaches. The form of this analysis is tight enough to provide several insights into the practical behavior of SGD+M, including suggesting hyper-parameter schemes and indicating why SGD+M is faster than SGD at the early stages of optimization. We believe our analysis technique is also useful in it's own right, and may be a good starting point for analyzing other methods that involve momentum.

There is a substantial body of prior work on the SGD+M method. Non-asymptotic convergence in the non-stochastic convex setting was first established by Ghadimi et al. (2015), where it is shown that for parameters of the form $\beta_k = k/(k+2)$ and $\alpha_k \propto 1/(k+2)$, the method obtains last iterate convergence rates comparable to gradient descent. They also show that when $\beta_k$ is constant the best convergence rate they are able to obtain is worse than gradient descent by a constant factor $\beta$. Unfortunately their proof technique does not extend readily to the stochastic setting. Flammarion and Bach (2015) consider both momentum and accelerated methods for convex quadratic problems, where they are able to establish bounds using the technique of difference equations, even with noisy (but not stochastic) gradients.

Yuan et al. (2016) analyze momentum methods under the assumption of strong convexity and small step sizes in the online setting, and show no actual advantage to momentum methods in this setting. Can et al. (2019) establish strong results in another special case, where gradient noise is bounded and the objective is either strongly convex or quadratic. Needell et al. 2014 also consider the strongly-convex case, using proof techniques developed for the randomized Kaczmarz algorithm. Also under a quadratic assumption, Jain et al. (2018) analyzed an accelerated scheme related to Nesterov's accelerated method in the stochastic case. While the heavy ball method is known to provide accelerated convergence rates for quadratic problems, these rates provably do not extend to the non-quadratic case (Kidambi et al., 2018).

Yan et al. (2018) provide the first analysis of momentum (with an earlier preprint Yang et al., 2016), including Nesterov's scheme, in the non-convex case, establishing a bound of the form:

$$\min_{k=0,\ldots,t} \mathbb{E}\left[\|\nabla f(x_k)\|^2\right]$$

$$\leq \frac{2\left[f(x_0) - f_*\right](1-\beta)}{t+1} \max\left\{\frac{2L}{1-\beta}, \frac{\sqrt{t+1}}{C}\right\} + \frac{CL\beta^2\left(G^2 + \sigma^2 + L\sigma^2\left(1-\beta\right)^2\right)}{\sqrt{t+1}\left(1-\beta\right)^3},$$

where $\|\nabla f(x)\| \leq G$, $\mathbb{E}\left[\|\nabla f(x,\xi) - \nabla f(x)\|^2\right] \leq \sigma^2$, $C$ is a positive constant, and f is $L$-Lipschitz smooth, for method Eq. 1. This rate is much looser than the rate we establish in this work, and our rate includes no unspecified constants. Yu et al. (2020) consider the distributed non-convex setting, where they establish a rate that is also looser than our own. A general result of almost-sure convergence is shown by Gadat et al. (2018) in the non-convex setting.

Recently, Sebbouh et al. (2020) establish rates for the convex and strongly convex settings in the stochastic case that mirror the tight rates in the deterministic case of Ghadimi et al. (2015), using a Lyapunov function analysis. Along with Tao et al., 2020 and Defazio and Gower, 2020, this line of work shows that the primary advantage of the heavy ball method over SGD is that it it is possible to show tight convergence of the last-iterate, rather than an average of iterates (as for SGD). Last-iterate convergence rates for SGD are weaker than the average iterate convergence unless very careful parameter schemes are used (Jain et al., 2019), and even then only when the stopping time is known in advance.

For the non-convex setting, the closest work to ours is that of Liu et al. (2020), who use a Lyapunov analysis and make use of the same $z_k$ quantity that we use in this work, as an ancillary point. In our view $z_k$ should be a key part of the algorithm, rather than a derived quantity. They give the following bound on their Lyapunov function $\Lambda_k$:

$$\mathbb{E}[\Lambda_{k+1}] - \Lambda_k$$

$$\leq \left(-\alpha + \frac{-\beta + \beta^2}{2(1-\beta)}L\alpha^2 + 4c_1\alpha^2\right)\mathbb{E}\left[\|g_k\|^2\right] + \frac{\beta^2}{2(1+\beta)}L\alpha^2\sigma^2 + \frac{1}{2}L\alpha^2\sigma^2 + 2c_1\frac{1-\beta}{1+\beta}\alpha^2\sigma^2$$

where $\Lambda_k = f(z_k) - f^* + \sum_{i=1}^{k-1} c_i \left\|x^{k+1-i} - x^{k-i}\right\|^2$. We refer the reader to their paper for details in the values of $c$, $\alpha$ and the settings in which this bound holds. This bound is looser than the one we derive, and provides less insight into the practical behavior of SGD+M than the bound we derive in this work. In other work on the non-convex case, Cutkosky and Mehta (2020) analyze a form of SGD+M with normalized steps. The recent work of Mai and Johansson (2020) analyze SGD+M under a weak convexity assumption as well as in the smooth case, using different proof techniques than we explore in this work, resulting in a looser bound.

## 2 The averaging form of momentum

The stochastic gradient method with momentum (SGD+M) is commonly written in the following form:

$$m_{k+1} = \beta_k m_k + \nabla f(x_k, \xi_k),$$
$$x_{k+1} = x_k - \alpha_k m_{k+1}, \tag{1}$$

where $x_k$ is the iterate sequence, and $m_k$ is the momentum buffer, and $\nabla f(x_k, \xi_k)$ the stochastic gradient at step $k$. For our analysis we will not use this form, instead, we will make use of the recently discovered averaging form of the momentum method (Defazio, 2019; Sebbouh et al., 2020), also discovered as a separate method (without relating to SGD+M) under the name SPA (stochastic primal averaging) by Tao et al. (2020):

$$z_{k+1} = z_k - \eta_k \nabla f(x_k, \xi_k),$$
$$x_{k+1} = (1 - c_{k+1}) x_k + c_{k+1} z_{k+1}.$$

Mapping SGD+M parameters to SPA parameters

(a) $\alpha_k$ decreased

(b) $\beta_k$ decreased

Figure 1: The behavior of the hyper-parameters of the SPA form when they are set so as to maintain an identical iterate sequence as the SGD+M form

For specific choices of values for the hyper-parameters, the $x_k$ sequence generated by this method will be identical to that of SGD+M. The quantity $z_k$ is actually used in some early analysis of momentum methods, but without this explicit transformation (Ghadimi et al., 2015). A continuous time version of this update is analyzed in Krichene et al. (2016), but without relating it to the heavy ball method.

The averaging form, compared to the standard form, appears to be easier to analyze theoretically, as the $z$ sequence arises naturally when performing a Lyapunov-style analysis of the method. The mapping between the two forms is described in the following theorem.

**Theorem 1.** *The $x_k$ sequences of the SPA method and SGD+M are equal when $z_0 = x_0$ and for all $k \geq 0$:*

$$\eta_{k+1} = \frac{\eta_k - \alpha_k}{\beta_{k+1}}, \qquad c_{k+1} = \frac{\alpha_k}{\eta_k},$$

*conversely, $\alpha_k = \eta_k c_{k+1}$, and $\beta_k = \frac{\eta_{k-1}}{\eta_k}(1 - c_k)$.*

This correspondence results in surprising dynamics when otherwise reasonable hyper-parameter schedules are mapped from one form to another. For illustration, we will consider the case where one or both of the parameters are changed by a fixed factor, as is commonly done when using a stage-wise schedule. We apply this change at step 20 of 100 steps, with $\beta = 0.9$ and $\alpha = 1.0$. Each case is shown in Figure 1.

**(a)** When the learning rate $\alpha$ of the SGD+M form is decreased by a fixed factor while $\beta$ is kept constant, the learning rate in the SPA form begins to grow geometrically, and $c$ shrinks geometrically. This is the most common schedule used in practice for the SGD+M method, and the fact that it causes such odd behavior in the SPA form is a cause for concern. This schedule in SPA form is NOT supported by our Lyapunov analysis.

**(b)** When the momentum constant $\beta$ is changed (in our example from 0.9 to 0.8), while keeping $\alpha$ constant, a similar geometric increase/decrease behavior occurs as in case 1.

Both behaviors above are unsatisfying when viewed from the perspective of the SPA method. We may also perform the reverse operation, and consider the behavior of the hyper-parameters of the SGD+M method when step-wise schedules are used for the SPA form (Figure 2).

**(a)** When $\eta_k$ is decreased 10 fold, a spike occurs in $\beta_k$, after which $\alpha_k$ drops 10 fold and $\beta_k$ drops back to it's earlier value.

Mapping SPA parameters to SGD+M parameters

(a) $\eta_k$ decreased

(b) $c_k$ increased

Figure 2: The behavior of the hyper-parameters of the SGD+M form when they are set so as to maintain an identical iterate sequence as the SGD+M form

**(b)** When $c_k$ is increased 10 fold, then the SGD+M form is better behaved, as $\alpha_k$ increases 10 fold and $\beta_k$ drops to 0. This is reasonable behavior as this change corresponds to removing the momentum in both forms, while attempting to keep the effective step size the same.

**(c)** As we show in Section 5, the most theoretically motivated choice is to actually change both $\eta_k$ and $c_k$. This unfortunately also results in a spike in $\alpha_k$

**(d)** Replacing the sudden change in $\eta_k$ and $c_k$ by a gradual change removes the spike and keeps $\beta_k$ below 1. We show in Section 6 that a gradual change is actually required by our Lyapunov theory.

## 3 Lyapunov analysis

In the Lyapunov analysis technique, a non-negative function $\Lambda_k = \Lambda(x_{0:k}, z_{0:k}, \dots)$ is defined in terms of all indexed quantities in the algorithm up to the current time-step, for the purposes of controlling the convergence of the optimization method under analysis. In the convex case, the standard approach is to show that $\mathbb{E}\left[f(x_k) - f_*\right] \leq \Lambda_k - \mathbb{E}\left[\Lambda_{k+1}\right] + \text{noise}$, after which we can apply a telescoping argument to complete the proof. In the non-convex case we instead attempt to control the norm of the gradient of $f$, through a bound of the form:

$$d_k \left\| \nabla f(x_k) \right\|^2 \leq \Lambda_k - \mathbb{E}\left[\Lambda_{k+1}\right] + \text{noise}$$

where $d_k$ is some constant, and with expectations over randomness in the current step $k$, conditional on prior steps (we use this convention in the remainder of this work). We call an equation of this form a Lyapunov step equation. In the case of SGD it is straight-forward to show that the Lyapunov step takes the following form, assuming $\mathbb{E}\left[\left\| \nabla f(x_k, \xi_k) \right\|^2\right] \leq G^2$) and that $f$ is $L$-Lipschitz smooth:

$$\frac{1}{\eta_k} \mathbb{E}\left[\left\| \nabla f(x_k) \right\|^2\right] \leq \Lambda_k - \mathbb{E}\left[\Lambda_{k+1}\right] + \frac{1}{2}LG^2 + R_k, \tag{2}$$

where $\Lambda_k = \eta_k^{-2} \mathbb{E}\left[f(x_k) - f_*\right]$ and $R_k = (\eta_k^{-2} - \eta_{k-1}^{-2})\left[f(z_k) - f_*\right]$. From this Lyapunov step equation, a standard telescoping argument (we give details in the appendix) completes the convergence rate proof, yielding a bound on $\mathbb{E}\left[\left\| \nabla f(x_i) \right\|^2\right]$ for a randomly sampled $i$.

### 3.1 Momentum case

In the appendix, we construct the following Lyapunov function $\Lambda$ for the SGD+M method in SPA form:

$$\Lambda_{k+1} = \frac{1}{\eta_k^2}\left[f(z_{k+1}) - f_*\right] + \frac{L}{\eta_k}\left(\frac{1}{c_k} - 1\right)\left[f(x_k) - f_*\right] + \frac{L}{2\eta_k^2 c_{k+1}^2}\left\|x_{k+1} - x_k\right\|^2 \tag{3}$$

**Theorem 2.** *The SPA method obeys the following Lyapunov step equation for $k \geq 1$, with expectations conditioning on $x_k$ and prior gradients $\nabla f(x_i)$ for $i \leq k$:*

$$\frac{1}{2\eta_k}\left\|\nabla f(x_k)\right\|^2 + \frac{1}{2\eta_k}\left\|\nabla f(z_k)\right\|^2$$

$$\leq \Lambda_k - \mathbb{E}\left[\Lambda_{k+1}\right] + L\mathbb{E}\left[\left\|\nabla f(x_k, \xi_k)\right\|^2\right] + R_k$$

$$+ \frac{1}{2}\left[\frac{1}{\eta_k^2}\left(\frac{1}{c_k} - 1 + \eta_k L\right)\left(\frac{1}{c_k} - 1\right) + \frac{1}{\eta_k}L\left(\frac{1}{c_k} - 1\right)^2 - \frac{1}{\eta_{k-1}^2 c_k^2}\right]L\left\|x_k - x_{k-1}\right\|^2. \tag{4}$$

*where the remainder term $R_k$ (active on steps where $\eta$ or $c$ changes) is defined as:*

$$R_k = \left[\frac{L}{\eta_k}\left(\frac{1}{c_k} - 1\right) - \frac{L}{\eta_{k-1}}\left(\frac{1}{c_{k-1}} - 1\right)\right]\left[f(x_{k-1}) - f_*\right] + \left[\frac{1}{\eta_k^2} - \frac{1}{\eta_{k-1}^2}\right]\left[f(z_k) - f_*\right]$$

This bound is our key theoretical result. We give the full telescoped proof using this bound in the appendix yielding a $\mathcal{O}(k^{-1/2})$ rate. The key differences between this bound and the bound for SGD (Equation 2) are:

1. The convergence rate is in terms of $\frac{1}{2\eta_k}\left\|\nabla f(z_k)\right\|^2 + \frac{1}{2\eta_k}\left\|\nabla f(x_k)\right\|^2$ for SGD+M compared to $\frac{1}{\eta_k}\left\|\nabla f(x_k)\right\|^2$ for SGD. When we telescope to give a convergence rate bound, the bound is on a randomly sampled iterate from a weighted set of $x_k$ and $z_k$ rather than just $x_k$.

2. There is an extra $\left\|x_k - x_{k-1}\right\|^2$ term on the right which will be negative and hence beneficial for typical choices of the hyper-parameters, as we show in Section 6.

3. The noise term $\mathbb{E}\left[\left\|\nabla f(x_k, \xi_k)\right\|^2\right]$ is weighted by $L$ for SGD+M and $\frac{1}{2}L$ for SGD. Although this noise term is twice as large for SGD+M, we show in Section 4, that almost half of it is canceled by the negative $\left\|x_k - x_{k-1}\right\|^2$ term when additional assumptions are made, meaning that the noise is actually essentially the same as SGD.

4. The Lyapunov function of SGD is just $\eta_k^{-2}f(x_k)$, whereas the Lyapunov function of SGD+M involves $\eta_k^{-2}f(z_k)$ plus two other terms. After telescoping for $T$ steps (as we show in the appendix), the $\left\|x_{k+1} - x_k\right\|^2$ term drops out, and the $\left[f(x_k) - f_*\right]$ term decays at a rate $\sqrt{T}$ faster than the other terms, making it negligible at the end of optimization for typical values of $c_k$, i.e. when $\left(\frac{1}{c_1} - 1\right) \ll \sqrt{T}$. These terms appear to be the main limiting factor for how small $c_k$ can be chosen (i.e. how much momentum is used).

5. The $R_k$ term is 0 when $\eta_k = \eta_{k-1}$ and $c_k = c_{k-1}$, otherwise it contains an "error" accumulated from changing the hyper-parameters. In a stage-wise hyper-parameter scheme this error accumulation happens only at the end of each stage, and it's contribution to the final convergence rate bound will be weighted with $1/T$, significantly smaller than the $1/\sqrt{T}$ weight of the primary terms. This is similar behavior to the $R_k$ term in the SGD step equation.

## 4 Insight #1: Momentum may cancel out noise during early iterations

The noise term in the Lyapunov step of SGD+M is twice as large as the noise term $\frac{1}{2}L\mathbb{E}\left[\left\|\nabla f(x_k, \xi_k)\right\|^2\right]$ in SGD. Although typically such small differences are disregarded in the analysis of optimization methods, in

this case we believe that this term gives substantial insight into the practical behavior of the two methods. The difference between the bounds on the convergence rate of the two methods will depend crucially on the magnitude of the negative $\|x_k - x_{k-1}\|^2$ term in comparison to this noise term. When this negative iterate difference term is sufficiently large, SGD+M can be expected to converge faster than SGD. In this section we analyze this term in detail. We will assume in this section that $c_k = c$ and $\eta_k = \eta$ are independent of $k$, we consider in Section 6 what happens to $\|x_k - x_{k-1}\|^2$ when they change in a step-wise scheme.

Firstly note that the the weight of $\|x_k - x_{k-1}\|^2$ in the Lyapunov step (4) can be written in the following form after expanding and simplifying when using constant hyper-parameters:

$$\frac{L}{2} \left[ -\frac{2}{\eta^2 c} + \frac{1}{\eta^2} + \frac{L}{\eta c^2} - \frac{L}{\eta c} \right].$$

To understand the magnitude of $\|x_k - x_{k-1}\|^2$, we may consider it's recursive expansion:

$$\|x_k - x_{k-1}\|^2 = (1-c)^2 \|x_{k-1} - x_{k-2}\|^2$$
$$+ c^2\eta^2 \|\nabla f(x_{k-1}, \xi_{k-1})\|^2 - 2\eta c^2 \left( \frac{1}{c} - 1 \right) \langle \nabla f(x_{k-1}), x_{k-1} - x_{k-2} \rangle. \tag{5}$$

This recursive expression may be further unwound, giving a geometrically decreasing weighted sequence. We consider the inner-product term in the next section, for the moment we assume that it has expectation zero. The gradient term $\|\nabla f(x_{k-1}, \xi_{k-1})\|^2$ here gives some insight into why we may expect cancelation against the noise term in the Lyapunov step. When this expression is unwound, it contains a contribution from all past gradients:

$$\sum_{i=0}^{k} (1-c)^{2i} c^2\eta^2 \mathbb{E} \left[ \|\nabla f(x_{k-i}, \xi_{k-i})\|^2 \right],$$

So the noise term $\frac{1}{2}L\mathbb{E} \left[ \|\nabla f(x_k, \xi_k)\|^2 \right]$ is not canceled immediately by the negative iterate distance $\|x_k - x_{k-1}\|^2$, instead, it cancels part of the noise from past iterations. In fact, we can see that after some step $i$, the noise term introduced by that step over and above SGD, namely $\frac{1}{2}L\mathbb{E} \left[ \|\nabla f(x_i, \xi_i)\|^2 \right]$ will be partially negated at every successive step, in a geometrically decaying fashion. Considering it as an infinite sum, we have:

$$\sum_{i=0}^{\infty} (1-c)^{2i} = \frac{1}{1-(1-c)^2} = \frac{1}{c(2-c)}$$

Is this sufficient for the negative terms to cancel the additional noise over SGD? Let's consider the weight heuristically before providing a more precise argument. Firstly, consider the weight in front of $\|x_k - x_{k-1}\|^2$. The dominating term in this expression for small $\eta$ and $c$ is $-L/\eta^2 c$. The $\|\nabla f(x_i, \xi_i)\|^2$ term is multiplied by $c^2\eta^2$ in the geometric sum. The infinite sum is above is $1/2c$ for small $c$, so we find that we have:

$$-\frac{L}{\eta^2 c} \cdot c^2\eta^2 \cdot \frac{1}{2c} = \frac{L}{2},$$

which is exactly large enough to cancel the additional noise. We can make this argument precise using the tools of Lyapunov analysis, without requiring the above simplifications. In particular, we can augment the Lyapunov function with an additional term:

$$\frac{L}{2\eta c^2} \left[ \frac{L(1-c)}{c(2-c)} - \frac{1}{\eta} \right] \|x_{k+1} - x_k\|^2.$$

As we shown in the appendix, as long as $\eta \le \frac{2c(2-c)}{L(1-c)}$, this term captures the additional noise introduced at each step $(k=i)$, and how it decays geometrically overtime. With the addition of this term in the Lyapunov function, the noise term reduces to

$$\left( 1 + \frac{\eta L(1-c)}{c(2-c)} \right) \frac{L}{2} \mathbb{E} \left[ \|\nabla f(x_i, \xi_i)\|^2 \right],$$

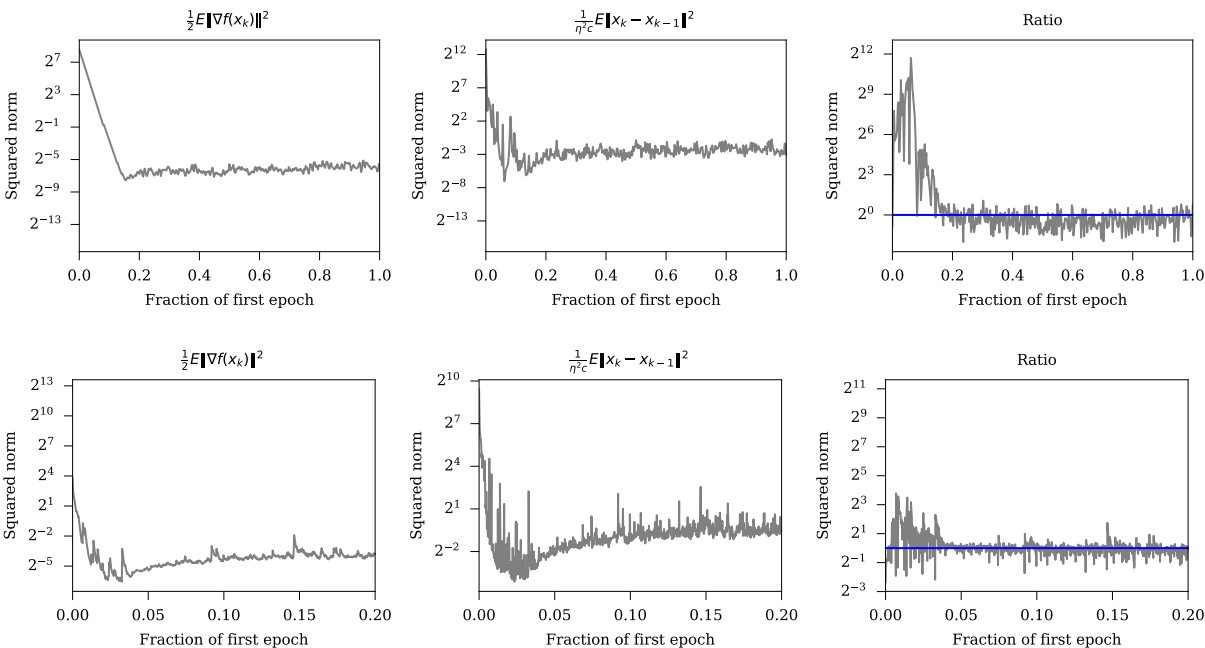

Figure 3: Quantities shown are during CIFAR10 (top row) and ImageNet (bottom row) training with momentum 0.9. Full details of the experimental setup are in the appendix. The extra negative $x_k - x_{k-1}$ term cancels out the large gradient norm squared term when the shown ratio (right) is above 1. Here this occurs for the initial steps during the first epoch of training.

almost matching SGD except for the term $\frac{\eta L(1-c)}{c(2-c)}$, which is very small for the $\eta \propto 1/(L\sqrt{T})$ values that the theory supports. Note however that by expanding $\|x_k - x_{k-1}\|^2$ we must also consider the additional inner-product terms introduced in Eq. 5, which we do in the next section.

**When momentum helps** By expanding the recursive definition of $\|x_k - x_{k-1}\|^2$, we have halved the noise term, but at the expense of introducing an inner-product term proportional to:

$$-2\eta c^2 \left(\frac{1}{c} - 1\right) \sum_{i=0}^{k} (1-c)^{2i} \langle \nabla f(x_i), x_i - x_{i-1} \rangle.$$

This term gives a precise characterization of when the convergence rate bound for SGD+M will be tighter than SGD; when for a particular weighted average, each $\nabla f(x_{i-1})$ is on average positively aligned or at worst orthogonal to the momentum buffer: $m_{i-1} \propto -(x_{i-1} - x_{i-2})$. If on average they are highly positively correlated, then we can expect momentum methods to significantly outperform non-momentum methods.

The correlation between the momentum buffer and the next gradient is not assured during optimization. Intuitively, a high correlation can be expected when the optimization path is heading in a steady direction, rather than oscillating around a minima or valley. This is particularly the case in the early stages of optimization, where there is a clear descent direction, in contrast to the later stages of optimization, where the optimization path will typically bounce around a minima or valley due to the noise introduced by using stochastic gradients. When the optimization path bounces around significantly, we would expect this inner-product term to be close to zero in expectation. So although the worst case behavior of SGD+M the convergence rate bound has double the noise of SGD, in practice we expect a behavior where at the early stages of optimization it may be faster, and at the later stages of optimization it will converge at the same rate as it enters a more noise dominated regime.

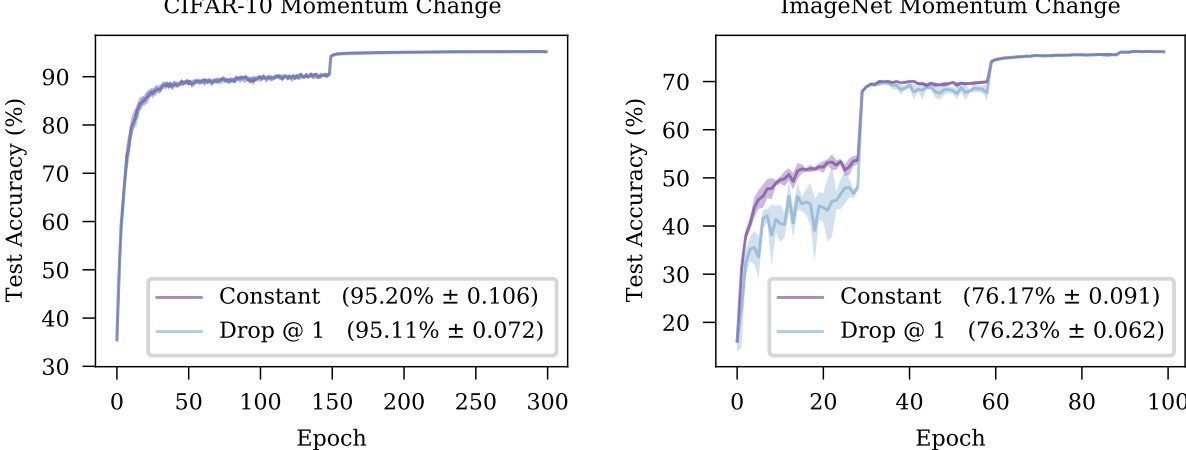

Figure 4: Removing momentum by setting $c_k = 0$ after the first epoch has no negative consequences on the final test accuracy for either problem. Experimental setup detailed in Section H.

**An empirical study** This result also suggests that momentum may ONLY be useful during the very earliest iterations. In the case of the CIFAR10 (Krizhevsky, 2009) problem shown, it appears to only provide a positive benefit for less than half of the first epoch, and the benefit is even shorter for ImageNet (Russakovsky et al., 2015). To test this hypothesis, we did a comparison where we turned off momentum after the first epoch. As shown in Figure 4, this gives the same test error curve and final test error as for when momentum is used for the whole run (two-sided t-test on the difference of means yields a p value of 0.61 for CIFAR10 and 0.705 for ImageNet).

Our theory suggests that we may directly measure when momentum is having a positive effect on convergence by comparing the expectations of the quantities $\frac{1}{\eta^2 c} \|x_k - x_{k-1}\|^2$ to $\frac{1}{2} \|\nabla f(x_k)\|^2$. Figure 3 shows the magnitudes of these two quantities (smoothed using an exponential moving average to approximate the expectation), as well as the ratio on two test problems. When considering the ratio, the $\|x_k - x_{k-1}\|^2$ term is significantly bigger at the earliest stages of optimization, and then quickly approaches the "noise" level of 1, corresponding to the inner-product discussed above being on average 0. Interestingly, the gradient norm is also very large during these early iterations, which may explain why momentum helps so much: It negates the contribution of the noise term to the convergence rate bound during the iterations when it is largest.

## 5 Insight #2: Reduce $c_k$ when you decrease $\eta_k$

Consider the remainder term $R_k$:

$$R_k = \left[ \frac{L}{\eta_k} \left( \frac{1}{c_k} - 1 \right) - \frac{L}{\eta_{k-1}} \left( \frac{1}{c_{k-1}} - 1 \right) \right] [f(x_{k-1}) - f_*] + \left[ \frac{1}{\eta_k^2} - \frac{1}{\eta_{k-1}^2} \right] [f(z_k) - f_*].$$

This term contains the additional error accumulated when the step size is changed. Our hyper-parameter choices should aim to keep this term small if possible. The second term involving $[f(z_k) - f_*]$ is exactly the remainder term that appears in SGD theory, and so we would not expect to be able to control it further. The first line involves both $c$ and $\eta$, and so we have a degree of control over it. We are particularly interested in stage-wise schemes, where at a certain time-step $T$ the step-size $\eta$ is divided by a factor $\phi$ (typically 10), i.e. $\eta_T = \eta_{T-1}/\phi$. In that case, we may keep the first term's coefficient at 0 if we choose parameters satisfying:

$$\frac{1}{c_T} = 1 + \frac{1}{\phi} \left( \frac{1}{c_{T-1}} - 1 \right).$$

For small $c$, this is approximately $c_T = \phi c_{T-1}$. I.e. when the step size is decreased by a factor $\phi$, we should increase $c$ by that same $\phi$ factor. Using the equivalence in Theorem 1, we can see that when constant step

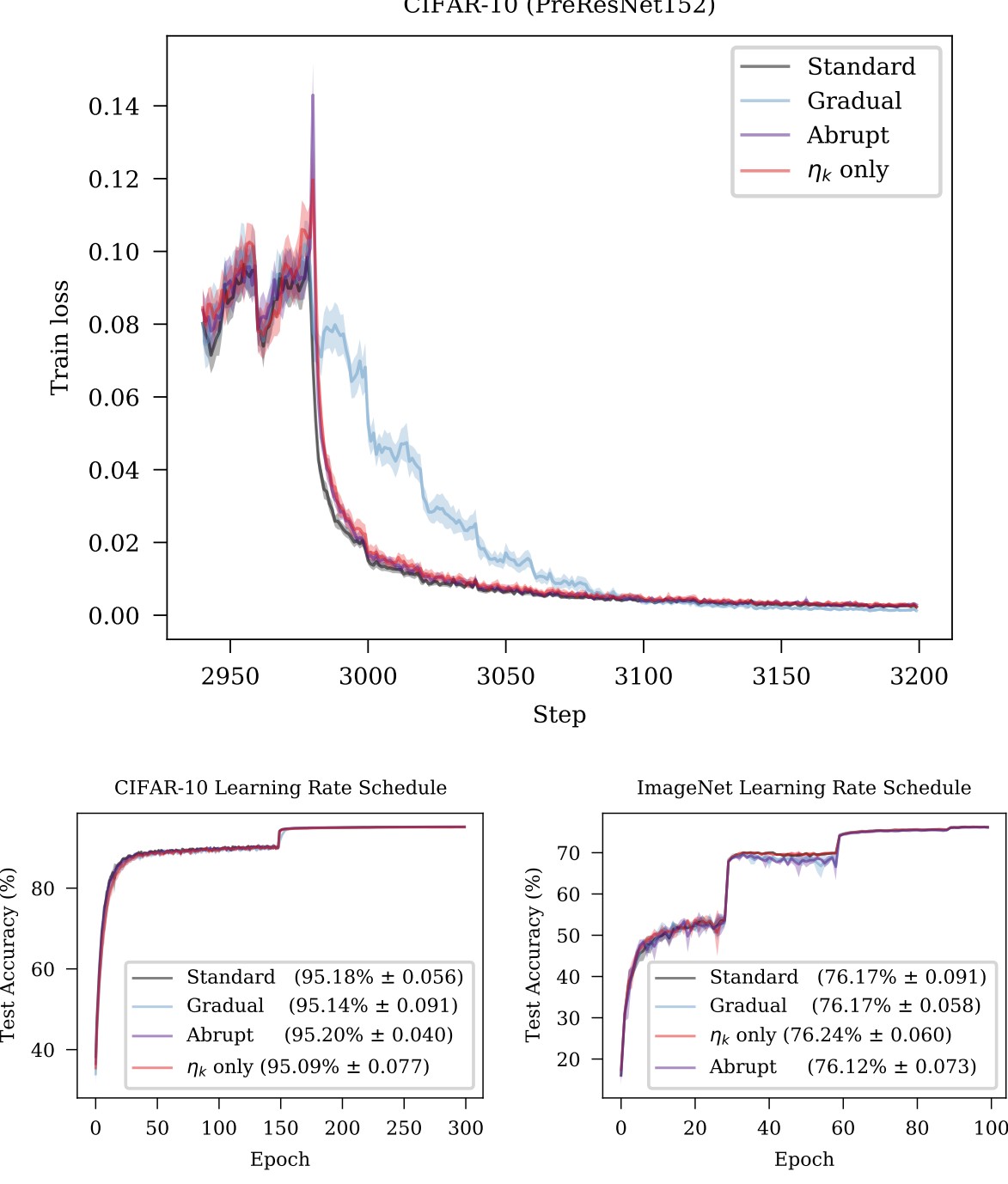

Figure 5: Top: training loss before and after then annealing point where the learning rate is decreased by a factor 10. Bottom: A comparison of the standard SGD+M schedule against the primal averaging abrupt and gradual schedules (Figure 6) and against an $\eta_k$ decrease only schedule. In each case the schedule is applied at the usual 10 fold LR decrease points, 150 and 225 epochs for CIFAR10, 30,60 & 90 epochs for ImageNet.

Mapping SPA parameters to SGD+M parameters under suggested scheme

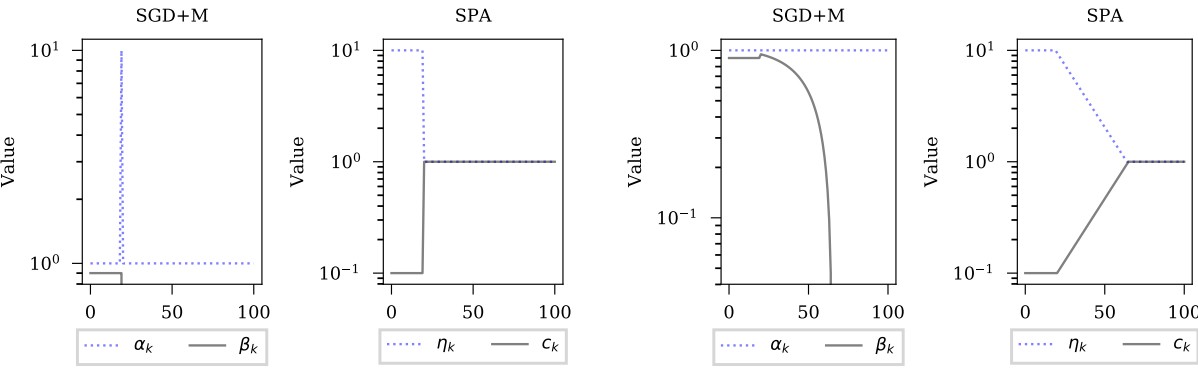

(a) $\alpha_k$ decreased and $c_k$ increased

(b) $\alpha_k$ decreased and $c_k$ increased gradually

Figure 6: The behavior of the hyper-parameters of the SGD+M form when they are set so as to maintain an identical iterate sequence as the SGD+M form, under a number of standard decrease schemes

sizes are used, the equivalence:

$$\beta = (1-c), \qquad \alpha = \eta c,$$

suggests that decreasing $\eta$ and increasing $c$ proportionally actually leaves the step size $\alpha$ the same, but decreases the amount of momentum $\beta$ in the SGD+M form. This suggests an alternative approach to the learning rate schedule, when working in SGD+M form: Decrease $\beta$ rather than decrease $\alpha$, up to the point where $\beta = 0$, corresponding to SGD without momentum.

Unfortunately, this scaling still presents problems, as we see in Figure 6 (left figure), there is an instantaneous spike in $\alpha_k$ when using this approach. Changing the learning rate by a large factor suddenly also effects the constants in front of the $\|x_k - x_{k-1}\|^2$ term in the Lyapunov step, resulting in this term being positive, rather than negative. We explore this difficulty and a potential solution in the next section (right figure).

## 6  Insight #3: Change hyper-parameters gradually

When constant momentum and step sizes are used, the weight of the term $\|x_k - x_{k-1}\|^2$ in the Lyapunov step is non-positive for values of $\eta$ larger than the typical $2/L$ maximum required for non-momentum methods:

$$\eta \leq \frac{2-c}{Lc(1-c)}. \tag{6}$$

However, when $\eta$ changes abruptly by large amounts between steps, this expression can not be satisfied. Instead, lets determine the largest multiplicative change in $\eta$ allowed between steps. Let $\eta_k = \eta_{k-1}/r$, where we expect $r$ to be larger than 1. We use $\eta$ to denote $\eta_{k-1}$ to simplify the notation. We also apply $\eta L \leq 1$ to simplify. This gives:

$$\frac{r^2}{\eta^2}\left(\frac{1}{c^2} - \frac{1}{c}\right) + \frac{r}{\eta}L\left(\frac{1}{c} - 1\right)^2 \leq \frac{1}{\eta^2 c^2},$$

Therefore $r^2 - r^2 c + r\eta L \left(1-c\right)^2 - 1 \leq 0$. Solving this quadratic equation gives two roots, one of which is always negative, the other root is:

$$r = \frac{-\eta L \left(1-c\right)^2 + \sqrt{\eta^2 L^2 \left(1-c\right)^4 + 4(1-c)c}}{2(1-c)}.$$

For instance with $c = 0.1$ $\eta L = 0.1$, a value of $r = 1.01$ satisfies the inequality. Note that when the learning rate is decreased further, the allowable values of $r$ increase. This suggests that at the point in which the

learning rate would normally decrease by a large factor such as 10 in a stage-wise schedule, instead the learning rate should be decreased geometrically, by a factor $\alpha$ each step, until it reaches the 10x lower value. Figure 6 shows that the $\alpha_k$ and $\beta_k$ values stay within reasonable ranges under this gradual scheme compared to the sudden spikes that are seen under other schemes. This provides further motivation for the use of a gradual reduction procedure.

**An empirical study**    The violation of the inequality that occurs when the learning rate is changed suddenly is **not just an artifact of the analysis** used, a spike in the training loss is readily observed in practice, An example that occurs during CIFAR-10 training is shown in Figure 5. Full details of the experimental setup are available in the Appendix. The gradual approach avoids the spike seen when the learning rate is changed suddenly. Although the training loss recovers rapidly after the spike, the gradual approach quickly obtains a lower training loss. The gradual approach modifies the standard scheme by increasing $c$ by 10-fold (up to a maximum of 1.0 for $c$) whenever $\eta$ is decreased 10-fold. Instead of an instantaneous change we changed both with a 1.0005 geometric factor each step until they reached their new value. As can be seen in Figure 5, there is also no loss of final test accuracy at all from using the gradual schedule for both CIFAR-10 or ImageNet. The result of applying two-sided t-tests for the difference of means pairwise between schedules shows no statistically significant differences at the 90% confidence level.

## Conclusion

Our analysis provides a better understanding of momentum methods for non-convex optimization through the lens of the primal averaging form. We characterize the extra terms introduced introduced into the Lyapunov analysis from the use of momentum, and show when these terms are beneficial and when they are harmful. We also analyze the behavior of the primal averaging form under changing step size schemes, and show the surprising result that standard schemes do not make sense in the averaging form, and suggest alternatives that are better behaved.

### Summary of Contributions

- We show that common learning rate schedules defined in the SGD+M form do not make sense when mapped to the SPA form.

- We provide a Lyapunov analysis of momentum in the non-convex setting which is tight enough to provide actionable insights into the behavior of momentum in practice.

- We provide insight into why SGD with momentum is typically no worse than SGD without momentum: the bound we derive for momentum has twice the noise term of SGD's bound, however when consecutive gradients are roughly uncorrelated or positively correlated the noise is almost halved, bringing it in line with SGD without momentum. We show this situation occurs in practice through an empirical investigation.

- We suggest fixes to the common stage-wise learning rate scheme that results in a schedule that makes sense in both SPA form and SGD+M form. We validate this schedule on CIFAR10 and ImageNet test problems. These fixes are directly motivated by minimizing our upper bound on the Lyapunov function.

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

# A    SGD+M and SPA equivalence

**Theorem 3.** *Define the SGD+M method by the two sequences:*

$$m_{k+1} = \beta_k m_k + \nabla f(x_k, \xi_k),$$
$$x_{k+1} = x_k - \alpha_k m_{k+1},$$

*and the SPA sequences as:*

$$z_{k+1} = z_k - \eta_k \nabla f(x_k, \xi_k),$$
$$x_{k+1} = (1 - c_{k+1}) x_k + c_{k+1} z_{k+1}.$$

*Consider the case where $m_0 = 0$ for SGD+M and and $z_0 = 0$ for SPA. Then if $c_1 = \alpha_0 / \eta_0$ and for $k \geq 0$*

$$\eta_{k+1} = \frac{\eta_k - \alpha_k}{\beta_{k+1}}, \qquad c_{k+1} = \frac{\alpha_k}{\eta_k},$$

*The x sequence produced by the SPA method is identical to the x sequence produced by the SGD+M method.*

*Proof.* Consider the base case where $x_0 = z_0$. Then for SGD+M:

$$m_1 = \nabla f(x_0, \xi_0)$$

$$\therefore x_1 = x_0 - \alpha_0 \nabla f(x_0, \xi_0) \tag{7}$$

and for the SPA form:

$$z_1 = x_0 - \eta_0 \nabla f(x_0, \xi_0)$$

$$x_1 = (1 - c_0) x_0 + c_0 (x_0 - \eta_0 \nabla f(x_0, \xi_0))$$
$$= x_0 - c_0 \eta_0 \nabla f(x_0, \xi_0) \tag{8}$$

Clearly Equation 7 is equivalent to Equation 8 when $\alpha_0 = c_0 \eta_0$.

Now consider $k > 0$. We will define $z_k$ in term of quantities in the SGD+M method, then show that with this definition, the step-to-step changes in $z$ correspond exactly to the SPA method. In particular, let:

$$z_k = x_k - \left( \frac{1}{c_k} - 1 \right) \alpha_{k-1} m_k. \tag{9}$$

Then

$$z_{k+1} = x_{k+1} - \left( \frac{1}{c_{k+1}} - 1 \right) \alpha_k m_{k+1}$$

$$= x_k - \alpha_k m_{k+1} - \left( \frac{1}{c_{k+1}} - 1 \right) \alpha_k m_{k+1}$$

$$= z_k + \left( \frac{1}{c_k} - 1 \right) \alpha_{k-1} m_k - \frac{\alpha_k}{c_{k+1}} (\beta_k m_k + \nabla f(x_k, \xi_k))$$

$$= z_k + \left[ \left( \frac{1}{c_k} - 1 \right) \alpha_{k-1} - \frac{\alpha_k}{c_{k+1}} \beta_k \right] m_k - \frac{\alpha_k}{c_{k+1}} \nabla f(x_k, \xi_k).$$

This is equivalent to the SPA step

$$z_{k+1} = z_k - \eta_k \nabla f(x_k, \xi_k),$$

as long as $\frac{\alpha_k}{c_{k+1}} = \eta_k$ and

$$0 = \left( \frac{1}{c_k} - 1 \right) \alpha_{k-1} - \frac{\alpha_k}{c_{k+1}} \beta_k$$
$$= (\eta_{k-1} - \alpha_{k-1}) - \eta_k \beta_k,$$

$$\text{i.e. } \eta_k = \frac{\eta_{k-1} - \alpha_{k-1}}{\beta_k}.$$

Using this definition of the $z$ sequence, we can rewrite the SGD+M $x$ sequence using a rearrangement of Equation 9:

$$m_{k+1} = \left(\frac{1}{c_{k+1}} - 1\right)^{-1} \alpha_k^{-1} \left(x_{k+1} - z_{k+1}\right),$$
$$= \frac{c_{k+1}}{1 - c_{k+1}} \alpha_k^{-1} \left(x_{k+1} - z_{k+1}\right),$$

as

$$\begin{aligned} x_{k+1} &= x_k - \alpha_k m_{k+1} \\ &= x_k - \frac{c_{k+1}}{1 - c_{k+1}} \left(x_{k+1} - z_{k+1}\right) \\ &= x_k - \frac{c_{k+1}}{1 - c_{k+1}} x_{k+1} + \frac{c_{k+1}}{1 - c_{k+1}} z_{k+1} \\ &= (1 - c_{k+1}) x_k + c_{k+1} z_{k+1}, \end{aligned}$$

matching the SPA update. $\qquad\square$

## B  Lemmas

**Lemma 1.** *(LEMMA 1.2.3, Nesterov (2013)) Suppose that $f$ is differentiable and has $L$-Lipschitz gradient:*

$$\|\nabla f(x) - \nabla f(y)\| \le L \|x - y\|, \tag{10}$$

*then:*

$$|f(y) - f(x) - \langle \nabla f(x), y - x \rangle| \le \frac{L}{2} \|x - y\|_2^2, \qquad \forall x, y \in \mathbb{R}^n. \tag{11}$$

*in particular,*

$$f(y) \le f(x) + \langle \nabla f(x), y - x \rangle + \frac{L}{2} \|x - y\|_2^2, \tag{12}$$

*and* $f(y) \ge f(x) + \langle \nabla f(x), y - x \rangle - \frac{L}{2} \|x - y\|_2^2. \tag{13}$

We will make heavy use of the fact that the $x_{k+1}$ update can be rearranged to give:

$$z_k = x_k - \left(\frac{1}{c_k} - 1\right) \left(x_{k-1} - x_k\right).$$

**Lemma 2.** *Suppose that $f$ is differentiable and has $L$-Lipschitz gradient, then the updates of the SPA form obey for $k \ge 1$:*

$$\begin{aligned} \frac{L}{c_{k+1}^2} \mathbb{E} \|x_{k+1} - x_k\|^2 &\le L \left(\frac{1}{c_k} - 1 + \eta_k L\right) \left(\frac{1}{c_k} - 1\right) \|x_k - x_{k-1}\|^2 + \eta_k^2 L \mathbb{E} \|\nabla f(x_k, \xi_k)\|^2 \\ &\quad + 2\eta_k L \left(\frac{1}{c_k} - 1\right) \left[f(x_{k-1}) - f(x_k)\right]. \end{aligned}$$

*Proof.* We may write the difference of the $x_k$ updates between steps as:

$$x_{k+1} - x_k = c_{k+1} \left(z_k - x_k\right) - \eta_k c_{k+1} \nabla f\left(x_k, \xi_k\right)$$

Recall that:

$$z_k - x_k = \left(\frac{1}{c_k} - 1\right) \left(x_k - x_{k-1}\right).$$

So:

$$x_{k+1} - x_k = c_{k+1}\left(\frac{1}{c_k} - 1\right)(x_k - x_{k-1}) - \eta_k c_{k+1}\nabla f(x_k, \xi_k)$$

Taking the squared norm and expanding, then taking expectations with respect to $\xi_k$ gives:

$$\mathbb{E}\|x_{k+1} - x_k\|^2 = c_{k+1}^2\left(\frac{1}{c_k} - 1\right)^2\|x_k - x_{k-1}\|^2 + c_{k+1}^2\eta_k^2\mathbb{E}\|\nabla f(x_k, \xi_k)\|^2$$
$$- 2\eta_k c_{k+1}^2\left(\frac{1}{c_k} - 1\right)\langle\nabla f(x_k), x_k - x_{k-1}\rangle$$

Now we apply the smoothness lower bound (Eq. 13):

$$f(x_{k-1}) \geq f(x_k) + \langle\nabla f(x_k), x_{k-1} - x_k\rangle - \frac{L}{2}\|x_k - x_{k-1}\|^2$$

Rearranged into the form:

$$-\langle\nabla f(x_k), x_k - x_{k-1}\rangle \leq f(x_{k-1}) - f(x_k) + \frac{L}{2}\|x_k - x_{k-1}\|^2$$

to give:

$$\mathbb{E}\|x_{k+1} - x_k\|^2 \leq c_{k+1}^2\left(\frac{1}{c_k} - 1\right)^2\|x_k - x_{k-1}\|^2 + c_{k+1}^2\eta_k^2\mathbb{E}\|\nabla f(x_k, \xi_k)\|^2$$
$$+ 2\eta_k c_{k+1}^2\left(\frac{1}{c_k} - 1\right)[f(x_{k-1}) - f(x_k)] + \eta_k L c_{k+1}^2\left(\frac{1}{c_k} - 1\right)\|x_k - x_{k-1}\|^2$$

Now group terms and multiply by $L/c_{k+1}^2$:

$$\frac{L}{c_{k+1}^2}\mathbb{E}\|x_{k+1} - x_k\|^2 \leq L\left(\frac{1}{c_k} - 1 + \eta_k L\right)\left(\frac{1}{c_k} - 1\right)\|x_k - x_{k-1}\|^2 + \eta_k^2 L\mathbb{E}\|\nabla f(x_k, \xi_k)\|^2$$
$$+ 2\eta_k L\left(\frac{1}{c_k} - 1\right)[f(x_{k-1}) - f(x_k)]$$

$$\square$$

**Lemma 3.** *Suppose that $f$ is differentiable and has L-Lipschitz gradients, then the updates of the SPA form obey for $k \geq 1$:*

$$\mathbb{E}[f(z_{k+1})] + \frac{\eta_k}{2}\|\nabla f(z_k)\|^2 \leq f(z_k) - \frac{\eta_k}{2}\|\nabla f(x_k)\|^2 + \frac{1}{2}\eta_k L^2\left(\frac{1}{c_k} - 1\right)^2\|x_k - x_{k-1}\|^2$$
$$+ \frac{1}{2}\eta_k^2 L\mathbb{E}\left[\|\nabla f(x_k, \xi_k)\|^2\right],$$

*where the expectation is with respect to $\xi_k$, and is conditional on the iterates and gradients from prior steps.*

*Proof.* Using $z_{k+1} = z_k - \eta_k\nabla f(x_k, \xi_k)$ and the smoothness upper bound (Equation 12):

$$\mathbb{E}[f(z_{k+1})] \leq f(z_k) - \eta_k\mathbb{E}\langle\nabla f(z_k), \nabla f(x_k, \xi_k)\rangle + \frac{1}{2}\eta_k^2 L\mathbb{E}\left[\|\nabla f(x_k, \xi_k)\|^2\right]$$
$$= f(z_k) - \eta_k\langle\nabla f(z_k), \nabla f(x_k)\rangle + \frac{1}{2}\eta_k^2 L\mathbb{E}\left[\|\nabla f(x_k, \xi_k)\|^2\right]$$
$$= f(z_k) + \frac{\eta_k}{2}\|\nabla f(z_k) - \nabla f(x_k)\|^2 - \frac{\eta_k}{2}\|\nabla f(z_k)\|^2 - \frac{\eta_k}{2}\|\nabla f(x_k)\|^2$$
$$+ \frac{1}{2}\eta_k^2 L\mathbb{E}\left[\|\nabla f(x_k, \xi_k)\|^2\right]$$

Now we use our assumption that the gradients are Lipschitz (Eq. 10):

$$\left\| \nabla f(z_k) - \nabla f(x_k) \right\|^2 \le L^2 \left\| z_k - x_k \right\|^2 = L^2 \left( \frac{1}{c_k} - 1 \right)^2 \left\| x_k - x_{k-1} \right\|^2$$

to give:

$$\mathbb{E}\left[ f(z_{k+1}) \right] + \frac{\eta_k}{2} \left\| \nabla f(z_k) \right\|^2 \le f(z_k) - \frac{\eta_k}{2} \left\| \nabla f(x_k) \right\|^2 + \frac{1}{2} \eta^k L^2 \left( \frac{1}{c_k} - 1 \right)^2 \left\| x_k - x_{k-1} \right\|^2$$
$$+ \frac{1}{2} \eta_k^2 L \mathbb{E}\left[ \left\| \nabla f(x_k, \xi_k) \right\|^2 \right]$$

$\square$

## C Proof of theorem 2

*Proof.* Consider Lemma 2 after taking expectations and dividing by $2\eta_k^2$:

$$\frac{L}{2\eta_k^2 c_{k+1}^2} \mathbb{E}\left[ \left\| x_{k+1} - x_k \right\|^2 \right]$$
$$\le \frac{1}{2\eta_k^2} L \left( \frac{1}{c_k} - 1 + \eta_k L \right) \left( \frac{1}{c_k} - 1 \right) \left\| x_k - x_{k-1} \right\|^2$$
$$+ \frac{1}{\eta_k} L \left( \frac{1}{c_k} - 1 \right) \left[ f(x_{k-1}) - f(x_k) \right]$$
$$+ \frac{1}{2} L \mathbb{E}\left[ \left\| \nabla f(x_k, \xi_k) \right\|^2 \right]$$

and Lemma 3 divided by $\eta_k^2$ :

$$\frac{1}{\eta_k^2} \mathbb{E}\left[ f(z_{k+1}) \right] + \frac{1}{2\eta_k} \left\| \nabla f(z_k) \right\|^2$$
$$\le \frac{1}{\eta_k^2} f(z_k) - \frac{1}{2\eta_k} \left\| \nabla f(x_k) \right\|^2$$
$$+ \frac{1}{2} \frac{1}{\eta_k} L^2 \left( \frac{1}{c_k} - 1 \right)^2 \left\| x_k - x_{k-1} \right\|^2$$
$$+ \frac{1}{2} L \mathbb{E}\left[ \left\| \nabla f(x_k, \xi_k) \right\|^2 \right]$$

Combining those bounds results in the following natural choice of Lyapunov function $\Lambda$:

$$\Lambda_{k+1} = \frac{1}{\eta_k^2} \left[ f(z_{k+1}) - f_* \right]$$
$$+ \frac{L}{\eta_k} \left( \frac{1}{c_k} - 1 \right) \left[ f(x_k) - f_* \right]$$
$$+ \frac{1}{2} L \frac{1}{\eta_k^2 c_{k+1}^2} \left\| x_{k+1} - x_k \right\|^2 \tag{14}$$

and yields the bound for $k \geq 1$:

$$
\begin{aligned}
\frac{1}{2\eta_k} & \|\nabla f(x_k)\|^2 + \frac{1}{2\eta_k} \|\nabla f(z_k)\|^2 \\
\leq {} & \Lambda_k - \mathbb{E}\left[\Lambda_{k+1}\right] + L\mathbb{E}\left[\|\nabla f(x_k, \xi_k)\|^2\right] \\
& + \frac{L}{2}\left[\frac{1}{\eta_k^2}\left(\frac{1}{c_k} - 1 + \eta_k L\right)\left(\frac{1}{c_k} - 1\right) + \frac{1}{\eta_k}L\left(\frac{1}{c_k} - 1\right)^2 - \frac{1}{\eta_{k-1}^2 c_k^2}\right]\|x_k - x_{k-1}\|^2 \\
& + \left[\frac{1}{\eta_k}L\left(\frac{1}{c_k} - 1\right) - \frac{1}{\eta_{k-1}}L\left(\frac{1}{c_{k-1}} - 1\right)\right][f(x_{k-1}) - f_*] \\
& + \left[\frac{1}{\eta_k^2} - \frac{1}{\eta_{k-1}^2}\right][f(z_k) - f_*]
\end{aligned}
\tag{15}
$$

$\square$

## D Telescoping

In order to complete a convergence rate proof, we must consider the behavior of the method at step 0. The above two lemmas are simplified in this case, yielding the following bound replacing Lemma 2:

$$
\mathbb{E}\left[\|x_1 - x_0\|^2\right] = c_1^2\eta_0^2\mathbb{E}\left[\|\nabla f(x_0, \xi_0)\|^2\right],
$$

and replacing Lemma 3

$$
\mathbb{E}\left[f(z_1)\right] + \frac{1}{2}\eta_0\|\nabla f(z_0)\|^2 \leq f(z_0) - \frac{1}{2}\eta_0\|\nabla f(x_0)\|^2 + \frac{1}{2}\eta_0^2 L\mathbb{E}\left[\|\nabla f(x_0, \xi_0)\|^2\right].
$$

Multiplying the first result by $L/(2c_1^2\eta_0^2)$ and dividing the second result by $\eta_0$, we may sum these equations to give:

$$
\begin{aligned}
\frac{1}{2\eta_0}\|\nabla f(x_0)\|^2 + \frac{1}{2\eta_0}\|\nabla f(z_0)\|^2 \leq {} & \frac{1}{\eta_0^2}[f(z_0) - f_*] - \frac{1}{\eta_0^2}\mathbb{E}[f(z_1) - f_*] \\
& - \frac{L}{2\eta_0^2 c_1^2}\mathbb{E}\left[\|x_1 - x_0\|^2\right] + L\mathbb{E}\|\nabla f(x_0, \xi_0)\|^2.
\end{aligned}
$$

Now consider the behavior of the SGD+M method when we use a fixed step size $\eta$. As long as

$$
\eta \leq \frac{2 - c}{Lc(1 - c)},
$$

and $E\left[\|\nabla f(x_k, \xi_k)\|^2\right] \leq G^2$, we may telescope from this base case to step $T$, yielding:

$$
\begin{aligned}
\frac{1}{\eta}\sum_k^T \mathbb{E} & \left[\frac{1}{2}\|\nabla f(x_k)\|^2 + \frac{1}{2}\|\nabla f(z_k)\|^2\right] \\
& \leq \frac{1}{\eta^2}[f(z_0) - f_*] + \frac{L}{\eta}\left(\frac{1}{c_1} - 1\right)[f(x_0) - f_*] + TLG^2.
\end{aligned}
\tag{16}
$$

Multiplying by $\eta/T$ gives a bound on the average iterate:

$$
\begin{aligned}
\frac{1}{T}\sum_k^T \mathbb{E} & \left[\frac{1}{2}\|\nabla f(x_k)\|^2 + \frac{1}{2}\|\nabla f(z_k)\|^2\right] \\
& \leq \frac{1}{\eta T}[f(z_0) - f_*] + \frac{L}{T}\left(\frac{1}{c_1} - 1\right)[f(x_0) - f_*] + \eta LG^2.
\end{aligned}
$$

Using the optimal step size $\eta^2 = T^{-1}L^{-1}G^{-2}\left[f(z_0) - f_*\right]$ gives:

$$\frac{1}{T}\sum_k^T \mathbb{E}\left[\frac{1}{2}\|\nabla f(x_k)\|^2 + \frac{1}{2}\|\nabla f(z_k)\|^2\right] \leq 2G\frac{\sqrt{L\left[f(z_0) - f_*\right]}}{\sqrt{T}} + \frac{L}{T}\left(\frac{1}{c_1} - 1\right)\left[f(x_0) - f_*\right],$$

whereas the more realistic step size $\eta^2 = T^{-1}L^{-2}$ gives

$$\frac{1}{T}\sum_k^T \mathbb{E}\left[\frac{1}{2}\|\nabla f(x_k)\|^2 + \frac{1}{2}\|\nabla f(z_k)\|^2\right]$$

$$\leq \frac{L}{\sqrt{T}}\left[f(z_0) - f_*\right] + \frac{L}{T}\left(\frac{1}{c_1} - 1\right)\left[f(x_0) - f_*\right] + \frac{G^2}{\sqrt{T}}.$$

In each case, the extra term $\left[f(x_0) - f_*\right]$ that differs from the standard SGD Lyapunov function decays at a $1/\text{T}$ rate, and so becomes negligible for large $T$.

## E    Removing the bounded gradients assumption

The above argument uses a bounded gradients assumption, however this assumption is easily replaced by a bound on the gradient variance:

$$\sigma^2 = \mathbb{E}\left[\|\nabla f(x_k, \xi_k) - \nabla f(x_k)\|^2\right].$$

Using $\mathbb{E}\left[\|\nabla f(x_k, \xi_k)\|^2\right] - \|\nabla f(x_k)\|^2 = \mathbb{E}\left[\|\nabla f(x_k, \xi_k) - \nabla f(x_k)\|^2\right]$ Equation 16 is replaced by

$$\frac{1}{\eta}\sum_k^T \mathbb{E}\left[\frac{1 - 2L\eta}{2}\|\nabla f(x_k)\|^2 + \frac{1}{2}\|\nabla f(z_k)\|^2\right]$$

$$\leq \frac{1}{\eta^2}\left[f(z_0) - f_*\right] + \frac{L}{\eta}\left(\frac{1}{c_1} - 1\right)\left[f(x_0) - f_*\right]$$

$$+ TL\sigma^2.$$

The key difference is that the factor of $\frac{1}{2}$ multiplying the squared gradient term is replaced by $\frac{1 - 2L\eta}{2}$. As long as $\eta \leq 1/(2L)$ this yields a comparable convergence bound as under the bounded gradient assumption, up to a factor of 2.

## F    SGD reference proof

We reproduce the standard argument for non-convex SGD convergence here using our notation for easy comparison to our SGD+M proof above. Consider the step $x_{k+1} = x_k - \eta_k \nabla f(x_k, \xi_k)$. Then:

$$f(x_{k+1}) \leq f(x_k) + \langle\nabla f(x_k), x_{k+1} - x_k\rangle + \frac{1}{2}L\|x_{k+1} - x_k\|^2$$

$$= f(x_k) - \eta_k\langle\nabla f(x_k), \nabla f(x_k, \xi_k)\rangle + \frac{1}{2}L\eta_k^2\|\nabla f(x_k, \xi_k)\|^2.$$

Taking expectations and using the bounded gradients assumption gives:

$$\mathbb{E}\left[f(x_{k+1})\right] \leq f(x_k) - \eta_k\|\nabla f(x_k)\|^2 + \frac{1}{2}L\eta_k^2 G^2.$$

Define $\Lambda_k = \eta_k^{-2}\mathbb{E}\left[f(x_k) - f_*\right]$: Then rearranging gives:

$$\frac{1}{\eta_k}\mathbb{E}\left[\|\nabla f(x_k)\|^2\right] \leq \Lambda_k - \mathbb{E}\left[\Lambda_{k+1}\right] + \frac{1}{2}LG^2 + (\eta_k^{-2} - \eta_{k-1}^{-2})\left[f(z_k) - f_*\right].$$

Assuming a fixed step size, we telescope from 0 to $T$ after taking total expectations:

$$\frac{1}{\eta} \sum_{k=0}^{T} \mathbb{E}\left[\|\nabla f(x_k)\|^2\right] \le \Lambda_0 - \mathbb{E}\left[\Lambda_{T+1}\right] + \frac{1}{2} L G^2 T.$$

So:

$$\frac{1}{T} \sum_{k=0}^{T} \mathbb{E}\left[\|\nabla f(x_k)\|^2\right] \le \frac{1}{\eta T} \left[f(x_0) - f_*\right] + \frac{1}{2} L \eta G^2,$$

using the optimal step size

$$\eta = \sqrt{\frac{2 \left[f(x_0) - f_*\right]}{T L G^2}}$$

gives:

$$\frac{1}{T} \sum_{k=0}^{T} \mathbb{E}\left[\|\nabla f(x_k)\|^2\right] \le \frac{G \sqrt{2L \left[f(x_0) - f_*\right]}}{\sqrt{T}},$$

which for large $T$, only differs from the SGD+M rate by a factor $\sqrt{2}$.

## G   Augmented Lyapunov

In Section 4, we consider the case of constant $\eta$ and $c$, and we introduce the additional assumption that $\langle \nabla f(x_{k-1}), x_{k-1} - x_{k-2} \rangle = 0$, so that:

$$\|x_k - x_{k-1}\|^2 = (1-c)^2 \|x_{k-1} - x_{k-2}\|^2 + c^2 \eta^2 \|\nabla f(x_{k-1}, \xi_{k-1})\|^2, \tag{17}$$

We want to modify the Lyapunov function so that we have:

$$\rho \Gamma_{k+1} \le \rho \Gamma_k + \rho_k c^2 \eta^2 \|\nabla f(x_k, \xi_k)\|^2,$$

where $\rho$ is a negative, and $\Gamma_{k+1} = \|x_{k+1} - x_k\|^2$. Consider the constants in front of the $\|x_k - x_{k-1}\|^2$ term in the Lyapunov step:

$$\frac{L}{2c\eta} \left[-\frac{2-c}{\eta} + \frac{L - Lc}{c}\right] \|x_{k+1} - x_k\|^2.$$

Using this expression, clearly our requirement on $\rho$ will be satisfied if:

$$\rho (1-c)^2 + \frac{L}{2c\eta} \left[-\frac{2-c}{\eta} + \frac{L - Lc}{c}\right] = \rho,$$

solving for $\rho$ gives:

$$\rho = \frac{L}{2\eta c^2} \left[\frac{L(1-c)}{c(2-c)} - \frac{1}{\eta}\right].$$

$\rho$ will be negative when:

$$\eta \le \frac{c(2-c)}{L(1-c)},$$

which covers all reasonable choices of hyper-parameters as considered in the convergence rate theory above. Using this $\rho$, we have an additional term in the Lyapunov step equation given by weighting the gradient noise term in Eq. 17 by $\rho$:

$$\rho c^2 \eta^2 \|\nabla f(x_k, \xi_k)\|^2 = \left[\frac{\eta L(1-c)}{c(2-c)} - 1\right] \frac{L}{2} \|\nabla f(x_k, \xi_k)\|^2.$$

This value is very close to $-\frac{L}{2} \|\nabla f(x_k, \xi_k)\|^2$ for sensible hyper-parameter values. For instance, for a typical $\eta = T^{-1/2} L^{-1}$ choice you get for the inner term:

$$\frac{\eta L(1-c)}{c(2-c)} - 1 = \frac{1-c}{\sqrt{T} c(2-c)} - 1,$$

which for $c = 0.1$ and $T = 10,000$, yields $\frac{1-c}{\sqrt{T} c(2-c)} - 1 = 0.047 - 1$.

# H    Details of experiments

In both cases below, when expressed in SPA form, the initial LR 0.1 corresponds to an initial learning rate of 1.0 and $c = 0.1$.

**CIFAR10**

Our data augmentation pipeline consisted of random horizontal flipping, then random crop to 32x32, then normalization by centering around (0.5, 0.5, 0.5). We used the standard learning rate schedule for this problem, consisting of a 10-fold decrease at epochs 150 and 225. Test/train/validate splits are standard. Total running time is < 24 hours per run. Our results are averaged over 20 seeds for each variant.

| Hyper-parameter | Value |
|---|---|
| Architecture | PreAct ResNet152 |
| Epochs | 300 |
| GPUs | 1xV100 |
| Batch Size per GPU | 128 |
| Decay | 0.0001 |

**ImageNet**

Data augmentation consisted of the RandomResizedCrop(224) operation in PyTorch, followed by RandomHorizontalFlip then normalization to mean=[0.485, 0.456, 0.406] and std=[0.229, 0.224, 0.225]. We used the standard learning rate schedule for this problem, where the learning rate is decreased 10 fold every 30 epochs. Test/train/validate splits are standard. Total running time is < 24 hours per run. Our results are averaged over 5 seeds for each variant.

| Hyper-parameter | Value |
|---|---|
| Architecture | ResNet50 |
| Epochs | 100 |
| GPUs | 8xV100 |
| Batch size per GPU | 32 |
| Decay | 0.0001 |

