# OpenReview forum: "Momentum via Primal Averaging: Theoretical Insights and Learning Rate Schedules for Non-Convex Optimization"
_TMLR — Rejected by TMLR_

### Review · Reviewer_QdVq · 2022-09-11

**Summary Of Contributions:**

This work studied SGD with momentum (SGD+M) from the perspective of primal averaging (following from Liu et al. 2020). The work makes use of a new Lyapunov function to derive a shaper convergence analysis for SGD+M.

Three insights are then discussed based on the obtained convergence bound:
 (1) momentum only helps in early stages of training, (2) the common stage-wise stepsize scheduler might be problematic, and (3) a fix to the stage-wise stepsize scheduler by gradually decreasing the stepsize.

Experiments on CIFAR-10 and ImageNet are provided to support the theory and insights.

**Requested Changes:**

Please see above.

**Strengths And Weaknesses:**

# Strengths
+ The paper is well-written and easy to read.
+ I especially like the illustration on how the hyperparameters for SGD+M correspond to the hyper parameters for SPA, and vice versa. These plots do a good job for providing intuitions on the two algorithms.
+ Theorem 2 is novel to my knowledge. And some of the insights drawn from Theorem 2 are interesting.


# Weakness
- I feel the theoretical contribution about this work is not clear based on current writing. In particular I feel the related works could be discussed with more technical details. For example, the authors have mentioned that their bound is sharper than [Yan et al. 2018] and [Liu et al. 2020] in introduction. However this is not explained rigorously. The bounds are quite complicated and the authors should explain which factors/terms in existing works are loose and in what sense.
- Insight #1 is somewhat sloppy in my perspective. If I understand correctly, the authors tried to compare the bounds for SGD+M and SGD to exam when momentum helps. The logic is fine but there are several issues in the conduction.
  - First of all the authors are comparing two upper bounds, this is unfortunate because one upper bound outperforms the other upper bound does not imply that the real performance of one algorithm outperforms the other.
  - Secondly, the arguments in Section 4 heavily replies on constants. However it is typical that upper bounds are sloppy in terms on constants. I am not sure if the "usefulness" claims on momentum are simply from the fact that one bound is less/more sloppier in constants than the other.
  - Moreover, there are other error terms in Eq. (4) (after doing telescope summation). I am not sure the noise term is the dominating term, especially in the early stage (the authors claim momentum only helps in early stage)

- Insight #2 is related to Insight #3. There are also sloppiness here and there. But my biggest concerns are from experimental results.
  - The authors write ".. and show the surprising result that standard schemes do not make sense in the averaging form, and suggest alternatives that are better behaved". But Experiments in Figure 5 clearly demonstrate that standard stepsize scheduler is not problematic in practice. Moreover, there are little difference between the final loss/accuracy obtained by standard stepsize scheduler and the proposed gradual stepsize scheduler. In my perspective, the experiments do not support the claimed benefits of the proposed method.
  - Moreover, even if the standard stage-wise stepsize scheduler for SGD-M can present numerical unstableness when converted to SPA, the stepsize scheduler might not be responsible for the issue. Because the authors did not show SGD-M with the stepszie scheduler is numerically unstable. In fact, as suggested by experiments, SGD-M with standard stepsize scheduler is sufficiently stable to reach a good solution. A more reasonable explaination is that SPA is not a suitable viewpoint for understanding SGD-M.

---

> ### Author Response · Authors · 2022-09-21
> **Rebuttal**
>
> Thank you for your insightful comments. We will address them individually.
>
> Regarding the general sloppiness of insight 1, we agree that it is far from rigorous. We present a fairly rigorous treatment via the use of an additional Lyapunov term in the Appendix section G, but even then this behavior is dependent on properties of the problem. We thought this argument was interesting enough to warrant discussion since it seems to correspond very closely to the empirical behavior we observe in Figure 3. This section could be considered more of a discussion after our main rigorous result in Theorem 2.
> The overall approach of comparing upper bounds as we do is only useful when the upper bounds are tight enough, which we believe to be the case here, as in both cases, SGD and SGD+M, a great deal of effort has been spent in getting them as tight a bound as possible. It's still not ideal, but realistically it's the best we can do on non-convex problems.
>
> Insight 2/3
> As we discuss in our rebuttal to Reviewer sCDv, the overall robustness of SGD on ResNet training to spikes like this does result in the classic step-wise scheme working well. We don't make the claim here that our gradual approach is better, just that it's more theoretically motivated and behaves just as good in practice as the stage-wise "cliff" schedule. ResNet training doesn't diverge even with much larger step-sizes, so it's really a poor test problem here, however we are not sure what standardized test problems are less robust, usually this is not a measured property noted in papers. We would be open to running additional comparisons here if you know of a particular problem where SGD (rather than Adam) is used and where convergence is particularly fragile.

---

### Review · Reviewer_mSMZ · 2022-09-13

**Summary Of Contributions:**

This submission studies step-size schedules for stochastic gradient descent
with Polyak momentum (SGD+M) through the lens of stochastic averaging.
By leveraging recent results showing the stochastic primal averaging
method is equivalent to SGD+M, the authors obtain an alternative, easier to analyze dynamical
system for which they derive a Lyapunov function.
Analyzing this function yields (1) a $O(1/\sqrt{T})$ convergence rate for SGD+M
under suitable parameter settings and (2) allows the author to study trade-offs
for different step-size schedules.
In particular, the authors show that the standard stage-wise decay schedule for
the step-size of SGD+M leads to unintuitive behavior of the primal averaging algorithm
and vice-versa.
Smooth decay schedules which do not cause parameter spikes in either space are proposed
as a solution.


**Broader Impact Concerns:**

None. This work is theoretical in nature.

**Requested Changes:**

As mentioned in "Strengths and Weaknesses" above, I suggest that the experiments in Figure 4 be revised and that additional experiments be performed to test the authors' hypothesis that momentum only helps at the start of optimization.
I think this is claim is the most interesting/salient aspect of the submission and that the manuscript would be greatly improved if additional, strong evidence could be included before publication.
Please see my remarks in the main review for specific recommendations.

**Strengths And Weaknesses:**

## Review

This is a polished submission well-suited to the TMLR format.
Although the convergence results presented are not ground-breaking, the analysis
is clean, succinct and will be useful for many members of the community.
The discussion of hyper-parameter tuning is interesting and sheds light on the empirical behavior of momentum methods.

### Correctness

I checked the proofs and the submission is theoretically sound as far as I am able to determine.
All theorems are supported by rigorous proofs which appear to be correct.
The experiments are well-executed and sufficient details are provided for replication.

#### Detailed Comments

**Figure 4**:
I don't think this experiment is sufficient to confirm the hypothesis that momentum is faster than SGD at the beginning of optimization.
I have several concerns here:
- I think it would be better to plot the (average) gradient norms with respect to $z_k$ and $x_k$, since the sum of these quantities is bounded by Theorem 2 and the subsequent analysis in Appendix D. We have no guarantees on the test accuracy, so it seems strange to compare on this quantity.
- The figure should also compare to SGD without momentum, since it could be that momentum has almost no effect on optimization performance.
- A two-sided t-test doesn't make sense since the test accuracy is non-negative.

I also suggest directly plotting the values of the inner-product term, since this is what (theoretically) controls the speed of optimization relative to SGD. Perhaps you can find some compelling evidence (and a sensible t-test) by plotting the behavior of this quantity over several trials.

**Figure 6**:
- I find this figure somewhat misleading. The full expression for $\beta_k$ according to Theorem 1 is

$$\beta_k = \frac{\eta_{k-1}}{\eta_k} (1 - c_k).$$

Suppose we decrease $\eta$ as $\eta_k = b \eta_{k-1}$, where $b < 1$, and increase $c$ as $c_{k} = c_{k-1} / b$.
This gives $\alpha_{k-1} = \eta_{k-1} c_{k} = \eta_{k-1} c_{k-1} / b$ and $\alpha_k = \eta_{k} c_{k+1} = \eta_{k-1} c_{k-1}$, so that $\alpha$ spikes at $k-1$ and returns to it's constant value at $k$.
However, we also have $\beta_k = 1/b ( 1 - c_{k-1} / b)$, which implies that $\beta$ spikes if $1 - c_{k-1} / b \neq 0$ before being constant at the value $\beta_k = (1 - c_{k-1} / b)$.
Figure 6 exactly choose $b$ so that $1 - c_{k-1} / b = 0$, which hides the fact that $\beta_k$ still spikes even when the two parameters are changed together.
Can the authors please address this issue?

**Section D**:
    - I believe you also require $c_{k-1} \leq c_k$ in order to drop the third term in Eq. 15 (RHS) before telescoping? It is not clear from the derivation whether $c$ is treated as a constant like $\eta$.


### Writing and Presentation:

The paper is well written and easy to follow.
I have several minor comments to improve the presentation:

- The introduction references undefined quantities several times.
    I suggest the authors either move Equation 1 and the definition of $z_k$ to
    Section 1, or move their literature review to a dedicated section later in the paper.
- The font sizes are too small for Figure 3 to be easily legible, especially when printed.
- Perhaps use a color other than gray in Figures 1,2,3,6. Including markers can also help color-blind readers.
- Figure 5 would be easier to interpret if the steps/epochs where the step-size changes were marked with a dashed vertical line.
- The conclusion should be its own numbered section.
- For completeness, I would state the conclusion of Appendix D as a theorem in the main paper.
- Consider grouping insights 1-3 as sub-sections of the same section.


#### Additional References:

You may find the recent work by Wang et al. [1] relevant, although it concerned with accelerated convergence
in the deterministic setting.

[1] Wang, Jun-Kun, et al. "Provable Acceleration of Heavy Ball beyond Quadratics for a Class of Polyak-Lojasiewicz Functions when the Non-Convexity is Averaged-Out." International Conference on Machine Learning. PMLR, 2022.


### Interest

Although SGD+M is a widely used optimization method, it is still poorly understand theoretically.
This provides further intuition for the behavior of momentum methods and will be interesting to many members of the research community.
The discussion of hyper-parameter optimization may also be appreciated by practitioners.

### Strengths and Weaknesses:

Strengths:
- The proof of convergence for SGD+M is surprisingly simple and succinct.
- The analysis of hyper-parameters for SGD+M via primal averaging is novel, potentially useful for practitioners,
and suggests more work is required to understand tuning for momentum methods.
- The text is well written.

Weaknesses:
- The most interesting claim (momentum only helps at the start of optimization) requires a more careful experimental setup and analysis.
- The presentation could be improved in some areas.

#### Minor Comments

Section 1:
    - What are the parameters $\alpha_k$ and $\beta_k$? These haven't been defined yet and so should not be referenced.
    - "f is L-Lipschitz smooth": "f" here should be written in math mode ("$f$").
    - What is $z_k$? Again, references to undefined quantities should be avoided.

Proof of Theorem 3:
    - You can use the `restatable` environment from `thmtools` to restate Theorem 1 instead of calling this Theorem 1.
    - As far as I can tell, there is no actual requirement for $z_0 = 0$, only that $z_0 = x_0$. Perhaps this is what the authors meant?
    - "zsequence" - > "$z$ sequence".

- Figure 4: "In the case of the CIFAR10 (Krizhevsky, 2009) problem shown, it appears to only provide a positive benefit for less than half of the first epoch, and the benefit is even shorter for ImageNet (Russakovsky et al., 2015)" --- This is referring to the behavior of the squared distance term (last column) in Figure 3, correct? Perhaps that link should be made directly for clarity.

---

> ### Author Response · Authors · 2022-09-21
> **Rebuttal**
>
> Thanks for the detail comments on presentation and notation - we will incorporate these changes. We address your major comments individually below:
>
> Figure 4) We agree with you and reviewer sCDv that we need to include a comparison to SGD without any momentum here as well. We will run the comparison for the camera ready. For the moment see Sutskever et. al (2013) for existing comparisons in the literature - using no momentum at all is much worse.
>
> Plotting true gradient norm of x and z, the actual convergence criterion used in our theory, is really problematic. We are using standard training pipelines for CIFAR10 and ImageNet that use data augmentation, so you can't just use full gradients to approximate the true gradient norm, even an entire epoch of gradients is not a noise free approximation due to the data augmentation.  The same problem occurs if we try and plot the inner product quantity.
>
> We believe the use of a two-sided test is reasonable here. Since the test accuracies are not close to 100% (i.e. their error bars do not approach the boundary of 100% accuracy), the behavior of the differences should be reasonably approximated by a t distribution. We would be happy to provide results using a different statistical test as well, do you have a suggestion for the approach we should take?
>
> Figure 6)
> This is a very good point. We choose the rate of change of the learning rate for this plot so that it satisfies the requirements set out in Section 6, which specifically prevents spikes. We will update the capture of the Figure to make it clear that we are doing this. It's absolutely the case that the hyper-parameters can still spike when using the SPA form if the condition in Section 6 is not followed.
>
> Section D) Yes we assume c is constant in this section for simplicity. We will add a comment clarifying this.

---

> > ### Comment · Reviewer_mSMZ · 2022-09-21
> > **Thanks for Clarifications**
> >
> > I'm glad we agree on many points. I'm looking forward to seeing the improved manuscript.
> >
> > As for plotting the true gradient norm of $x$ and $z$: I don't see the need to perform these experiments on ImageNet and CIFAR10 if the training pipelines cause extra difficulties. The claim in the paper is made for the general setting of stochastic, non-convex optimization, so why not consider a setting where you have more control over the optimization procedure> For example, shallow ReLU networks on MNIST would be a fine starting place for investigating this problem.
> >
> > The choice of statistical test is somewhat annoying. I would suggest using bootstrapped confidence intervals, but the difference in sample means is still bounded between $-1$ and $1$. I suspect the formal conditions for the bootstrap (e.g. CLT-like behavior of the mean difference) are not guaranteed arbitrarily close to the boundaries. This is why I suggested testing the inner-products terms---this quantity is unbounded and poses less of an issue for testing. With all that said, I don't think this is a critical issue.

---

> > > ### Author Response · Authors · 2022-09-26
> > > **Inner product plot**
> > >
> > > Yes, the argument for plotting the inner-product as well is compelling. We will add that plot for the camera ready. We will investigate to what degree we can produce a realistic plot of the gradient norm as well, hopefully we can include something in the camera ready, we need to measure the relative error empirically to get an idea of if it is feasible. If we can't do it for CIFAR10 we will look at a smaller network on MNIST.

---

### Review · Reviewer_sCDv · 2022-09-13

**Summary Of Contributions:**

This paper provides an analysis of a form of SGD with momentum through a particular Lyapunov potential. This recovers the standard $\E[\|\nabla f(x)\|]\le O(1/k^{1/4})$ critical-point convergence result. The key contribution is to observe a correspondence between different expressions for momentum (called the SGD+M and SPA forms in the paper), and investigate the effects of different learning rate or momentum parameter schedules. In particular, it is observed that intuitively reasonable schedules (e.g. step decay of learning rates) correspond to less intuitive schedules in the other form. To complement this intuitive picture, sharp decays of the learning rate appear to cause certain terms in the Lyapunov decrement to become transiently large, and produce a clear spike in error on CIFAR10.
Specifically, the Lyapunov analysis produces something similar to:
$$
\mathbb{E}[\|\nabla f(x_k)\|^2/\eta_k + \|\nabla f(z_k)\|^2/\eta_k] \le \mathbb{E}[\Lambda_k - \Lambda_{k+1}  + L|\nabla f(x_k, \xi_k)\|^2] +R_k  +A_k\|x_k-x_{k-1}\|^2]
$$
Where $R_k$ is a somewhat involved “remainder term” that is zero if the learning rate and momentum parameter are not changing, and $A_k$ is a somewhat involved term that is expected to be negative. However, when the learning rate is decayed too aggressively, $A_k$ can become positive, motivating a change to a smoother decay that does not cause any large terms in the Lyapunov analysis. Empirically, this change eliminated the transient increase in the error. Further, when $A_k$ is negative, we would like to have a larger $\|x_k-x_{k-1}\|$, which intuitively will happen  when the gradients are frequently pointing in similar directions so that momentum can accumulate (i.e. far from a local minimum). This intuition is supported by experiments showing that on CIFAR10 and ImageNet, turning off momentum after one epoch appears to not decrease final accuracy.

**Broader Impact Concerns:**

I have no concerns

**Requested Changes:**


A control on figure 4 with no momentum is necessary to strengthen the argument that momentum for one epoch is useful.



**Strengths And Weaknesses:**

This paper provides some interesting and perhaps surprising observations about different forms of momentum. The explanations are well motivated and the paper reads well. I like the idea of making changes to learning rate schedules based on fine-grained analysis of the Lyapunov function, and I can see how some of the high-level ideas might suggest future optimization methods.

My main concern is that the experiments don’t seem to really show that the proposed changes actually help in any way - at best they do not hurt in the end, but in some cases they seem to actually be detrimental for most of the optimization procedure.

For example, the observation that one can turn off momentum after one epoch is justified by an experiment in which the optimizer that turns off momentum seems to be worse than the one with momentum for almost the entire training time, and only at the end do they converge to the same error. This makes me wonder: is even one epoch of momentum necessary? What if we just didn’t use momentum? What if we used momentum for the first 0.2 fraction of the epoch as suggested by Figure 3? I think this point that momentum is helpful for the first epoch would have been much strengthened by such controls. Further, in figure 3, the gradient norm is measured, but actually I would expect large gradient norm to be *helpful* for optimization (at least in theory for smooth objectives as studied here).  It is a large *variance* that is bad. Does figure 3 measure the $\mathbb{E}[\|\nabla f(x_k)\|^2]$ as written, or is it actually measuring $\mathbb{E}[\|\nabla f(x_k, \xi_k)\|^2]$? In the former case, larger values seems actually beneficial, while in the latter case the variance is confounded with the signal. It would be better to measure something closer to the gradient variance if possible.

Further, while the observation that reasonable learning rate schedules for SGD+M result in strange updates for SPA is intriguing, I am not convinced that this should be such a drawback: why is it so bad to have large transient spikes in the learning rates for SPA? The Lyapunov-motivated alteration to the learning rate that does not produce positive terms in the step equation does remove the transient error increase in figure 5, but a cost of significantly slowing down the optimization since the transient increase is truly transient and is almost immediately recovered. This seems mysterious: is the transient spike actually helpful in some way in the medium term?


Overall, this paper does a good job of presenting several intriguing observations about the dynamics of momentum methods - indeed, my main concern is essentially that the paper raises more questions than it answers. In this way, I could see it being of some interest as some motivation for future investigation.

---

> ### Author Response · Authors · 2022-09-21
> **Rebuttal**
>
> Thanks for the detailed comments!
>
> Regarding Figure 4, the lower observed test loss at the early stages of optimization is really interesting. Our opinion is that the variance in the convergence of the drop@1 method is much higher, resulting in highly variable test accuracy, while the expectation is still making the same steady progress as without momentum. We didn't investigate this, but it could easily be confirmed by plotting the test accuracy of an averaged iterate rather than the latest iterate, if we use an exponential-moving-average with parameter 0.9, it may behave similarly to SGD+M. We will run this experiment for the Camera ready.
>
> Note that training without momentum at all gives drastically worse results, this has been observed in the literature before (Sutskever et. al 2013), but we agree it would be good to show it here as well. We will update this plot for the camera ready with a third line showing the learning curves without any momentum.
>
> For our measurements of gradient norm in Figure 3, we plot an exponential moving average of the batch gradient, not the expected gradient, so it does include noise. Our belief is that the noise is relatively low at these early stages of optimization. We will clarify this in the paper, currently it is misleading.
>
> Regarding the transient spikes observed with standard SGD schedules - it's really hard to say how important they are just from empirics. ResNet stage-wise training has significant selection bias - if the spikes caused serious problems, then this schedule would not have become so widely used. However it may well be that on problems that are less robust during training that more significant problems could occur. Given our focus on theory in this paper, we didn't dedicate significant resources to finding other less robust problems to test on, but rather decided to focus on key standardized test problems.

---

### Decision · Action_Editors · 2022-10-25

**Recommendation:** Reject

**Comment:**

The reviewer praised that (a) this paper provides some interesting and perhaps surprising observations about different forms of momentum, (b) that the paper is well-written and easy to follow, and (c) a great fit for the TMLR audience.

Among the main concerns of the reviewers were that the experimental claim could be strengthened by further ablation studies and reporting additional quantities, but also that the _necessity_ to avoid spikes has not been clearly demonstrated. (I acknowledge that such a demonstration may be beyond the scope of this paper and might not be necessary, but then some of the claims should perhaps be weakened or revised.)

Although the authors promised to address a most of these concerns in the rebuttal, the authors did not yet submit a revision. In the internal discussion, two reviewers argued that they wished that these edits would go through some additional reviewing phase and not just be added to the camera-ready version. That is why I decided to reject this version, with strong encouragement to resubmit to TMLR.

Among other points raised by reviewers (which could be considered in the resubmission), are:
- more complete & careful exploration of the phenomena that momentum only helps in the early iterations
- a few clarifying/technical comments in the comparisons to related works (in particular, a bit more detailed (or accessible) explanations for the comparisons to [Yan et al. 2018] and [Liu et al. 2020])


**Audience:**

Yes, the results and findings are of interest to the TMLR audience.

**Claims And Evidence:**

This paper studies step-size schedules for stochastic gradient descent with Polyak momentum (SGD+M). By utilizing an equivalent rewriting of the method as stochastic primal averaging (SPA), the paper explains that the commonly used stage-wise schedule in SDG+M leads to unintuitive behavior of SPA and proposes smooth decay schedules as an alternative.

The reviewers judged the _theoretical_ contributions as sound, supported by proofs. (Reviewer QdVq noted that a few statements in the main section lack rigor, or should be marked a bit more clearly as discussion -- but these suggestions look easy to address and do not impact the main contribution & decision).

However, all reviewers argued that the _experimental_ findings do not fully support all claims in the paper and some clarifications (or additional plots) are needed.

Furthermore, there was consensus among the reviewers that the claim "spikes [in the learning rate for SPA] should be avoided" is not yet corroborated by strong evidence, neither in theory nor experiments.